# Facile Synthesis and Redox Behavior of an Overcrowded Spirogermabifluorene

**Shogo Morisako** [1,2], **Kohsuke Noro** [3] **and Takahiro Sasamori** [1,2,3,*,†]

1   Division of Chemistry, Faculty of Pure and Applied Sciences, University of Tsukuba, 1-1-1 Tennoudai, Tsukuba 305-8571, Ibaraki, Japan; morisako.shogo.gf@u.tsukuba.ac.jp
2   Tsukuba Research Center for Energy Materials Sciences (TREMS), University of Tsukuba, Tsukuba 305-8571, Ibaraki, Japan
3   Graduate School of Science, Nagoya City University, Yamanohata 1, Mizuho-cho, Mizuho-ku, Nagoya 467-8501, Aichi, Japan; kou.n.ty1225@gmail.com
*   Correspondence: sasamori@chem.tsukuba.ac.jp; Tel.: +81-29-853-4412
†   Dedicated to Toshiaki Noda on the occasion of his 75th birthday.

**Abstract:** A spirogermabifluorene that bears sterically demanding 3,3′,5,5′-tetra(*t*-butyl)-2,2′- biphenylene groups (**1**) was obtained from the reaction of in-situ-generated 2,2′-dilithiobiphenylene with $GeCl_2 \cdot (dioxane)$. The solid-state structure and the redox behavior of **1** were examined by single-crystal X-ray diffraction analysis and electrochemical measurements, respectively. The sterically hindered biphenyl ligands endow **1** with high redox stability and increased electron affinity. The experimental observations were corroborated by theoretical DFT calculations.

**Keywords:** spirobifluorene; spirobigermabifluorene; spiroconjugation; germanium; biphenyl ligand; redox behavior; DFT calculations

## 1. Introduction

With the aim of constructing redox-active molecules, biphenyl ligands are very interesting π-organic bidentate ligands that exhibit redox activity on account of the effectively conjugated benzene rings [1–3]. Spirobifluorenes (**I**) that contain two mutually perpendicular biphenylene π-frameworks on a central tetravalent carbon atom are potentially efficient optoelectronic materials due to their unique photophysical properties derived from the spiroconjugation (Figure 1) [4–10]. Two main methods are conceivable for the modification of the properties of spirobifluorenes: (i) functionalization on the periphery of the biphenyl skeletons and (ii) replacement of the central element, which creates heterospirobifluorenes. Especially the replacement of the central carbon atom with heavier main-group elements should be interesting due to the characteristic low-lying LUMO that originates from the spiroconjugation through the σ-bonding of the central heavier element [11–16]. For example, a spirosilabifluorene (**II**), which contains a central silicon atom instead of the central carbon atom in the spirobifluorene skeleton, has attracted much attention due to its low-lying LUMO and small HOMO-LUMO gap, which affords effective photoabsorption and electron-transporting properties [11–19]. A vital requirement for such molecules as electron-transporting materials is redox stability, i.e., the anionic/cationic species generated via one-electron reduction/oxidation should exhibit relatively high stability. In this context, a heterospirobifluorene has already shown electrochemical stability upon reduction [11–19].

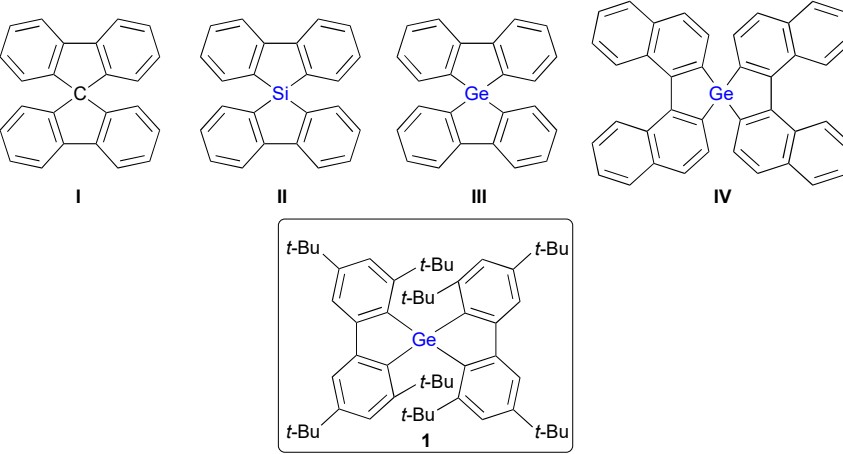

**Figure 1.** Spirobifluorene (**I**) and its Si- (**II**) and Ge-analogues (**III**, **IV** and **1**).

Spirogermabifluorenes (**III**) are expected to exhibit even lower LUMO levels relative to those of spirosilabifluorenes. While stable spirogermabifluorenes have already been synthesized and structurally characterized, spirogermabifluorenes are, in contrast to spirosilabifluorenes, unstable to chemical reduction, which results in the decomposition of the spirogermabifluorene skeleton [20]. One way to increase the redox stability of spirogermabifluorenes is using sterically demanding substituents that are able to surround the strained central atom to avoid intramolecular subreactions and the subsequent ring-opening [21–24]. However, attaching two biphenylene units that bear bulky substituents to a tetravalent germanium center via nucleophilic substitution can be expected to be difficult on account of steric congestion. For example, the reaction of 2,2-dilithio-1,1′-binaphtyl with GeCl$_4$ afforded the corresponding binaphtyl-substituted spirogermabifluorene (**IV**) in only very low yield (1.8%) [25]. In this article, we report an alternative synthetic route to an overcrowded spirogermabifluorene that bears 3,3′,5,5′-tetra-*t*-butylbiphenyl ligands (**1**), using GeCl$_2$·(dioxane) as the germanium source.

## 2. Results and Discussion

2,2′-Dibromo-3,3′,5,5′-tetra-*t*-butylbiphenyl (**3**) was prepared via the bromination of biphenyl **2** [26]. When **3** was treated with *n*-BuLi (3.0 e.q.) at −80 °C in Et$_2$O, the color of the solution changed to yellow, indicating the generation of 2,2′-dilithio-3,3′,5,5′-tetra-*t*-butylbiphenyl (**4**), the quantitative formation of which was confirmed by a trapping reaction using MeI [27]. Addition of a solution of the in-situ-generated **4** to a solution of GeCl$_4$ (5.0 eq), followed by the addition of an excess of LiCl, resulted in the quantitative formation of dichlorogermane **5** (isolated yield: 61%), which was characterized spectroscopically and structurally (vide infra). In the expectation of the formation of the overcrowded spirogermabifluorene **1**, we attempted the reaction of **5** with in-situ-generated **4**. However, only a complicated mixture including biphenyl **2** was obtained, and evidence for the formation of the expected spirogermabifluorene **1** was not observed (Scheme 1). The attempted synthesis of **1** by treatment of **4** with 0.5 equivalent of GeCl$_4$ was also unsuccessful, giving only a complicated mixture including **2** and **5**. Most likely, the steric hindrance of dichlorogermane **5** inhibits a further nucleophilic attack on the germanium atom.

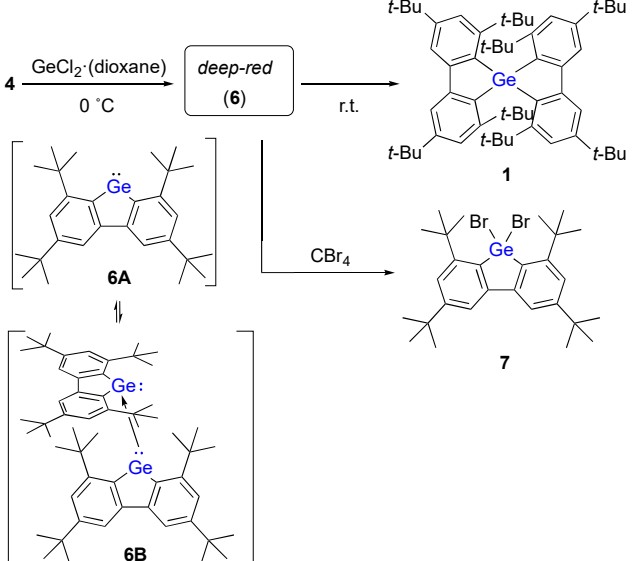

**Scheme 1.** (**a**) Synthesis of dichlorogermane **5**. (**b**) Attempted synthesis of **1** using conventional substitution reactions with Ge$^{IV}$ species.

Conversely, when an equimolar amount of GeCl$_2$·(dioxane) was added to an Et$_2$O solution of **4** at −80 °C, a deep-red suspension was obtained. When the mixture was warmed to r.t., its color changed to orange, and spirogermabifluorene **1** was obtained in 22% isolated yield after purification by gel-permeation chromatography (GPC; eluent: toluene) (Scheme 2). In order to identify possible intermediates, UV/vis spectra were recorded at low temperature of the deep-red ether solution and of the colorless solution obtained after exposure of this ether solution to air. The differential spectrum of the two spectra showed a characteristic absorption at $\lambda_{max}$ = 360 nm, indicating the formation of the corresponding Ge(II) species (**6**), i.e., germylene **6A** or digermene **6B**, as intermediates of the reaction (Scheme 2) [28]. Addition of CBr$_4$ to the deep-red ether solution furnished the corresponding dibromogermane **7** [29], which corroborates the anticipated presence of **6A** or **6B** as an intermediate.

**Scheme 2.** Synthesis of spirogermabifluorene **1** and dibromogermane **7**.

Although the mechanism of the reaction of **4** with GeCl$_2$·(dioxane) to furnish spirogermabifluorene **1** is not clear in all details at present, the formation of **1** can potentially be explained in terms of a disproportionate reaction of germylene-dimer **6B**, i.e., from Ge(II) to Ge(IV) and

Ge(0). On the basis of DFT calculations at the B3PW91-D3(BJ)/6-311G(2d,p) level of theory for the real models, a plausible reaction pathway is proposed in Scheme 3 [30].

**Scheme 3.** Plausible mechanism for the formation of spirogermabifluorene **1**. Calculated at the B3PW91-D3(BJ)/6-311G(2d,p) level of theory.

Germylene **6A** can form its dimer **6B** with an energy gain of 21.4 kcal/mol, which was calculated for the real models at the B3PW91-D3(BJ)/6-311G(2d,p) level of theory. Thus, a potential-energy-surface (PES) search was carried out starting from germylene dimer **6B**. On the calculated PES, the carbon migration could occur to give the corresponding germylgermylene (**8**) with a barrier of $\Delta G^{\ddagger}$ = 24.0 kcal/mol, which is comparable to a case of aryl migration of a tetra-aryldigermene that furnished the corresponding germyl-germylene [31]. Then, a further carbon migration in **8** could form the spirogermabifluorene skeleton bearing "Ge" on the aryl ring (**9**) ($\Delta G^{\ddagger}$ = 40.4 kcal/mol). Although the barrier from **8** to **9** seems to be a bit higher than expected based on the experimental results, where the reaction could occur below room temperature, the solvent effect and/or the coordination of the contaminated chloride ions in the real situation could promote the carbon-migration processes, thus lowering the reaction barrier. In other words, the release of Ge(0) from **9** to give **1** could proceed smoothly due to the gain of aromatic stabilization of the biphenyl ligand, while the process could be difficult to calculate due to the unclear situation with respect to Ge(0) being eliminated from **9**. In any case, the biphenyl-substituted germylene can be considered an appropriate precursor for a spirogermabifluorene.

The molecular structures of spirogermabifluorene **1** and dichlorogermane **5** determined by single-crystal X-ray diffraction analysis are shown in Figure 2 [32]. The Ge–C(biphenyl) bonds in **1** (1.974(5), 1.969(5), 1.972(5), 1.971(5) Å) are slightly longer than those of less hindered spirogermabifluorene **III** (1.938(2), 1.945(2) Å) [20] and **5** (ca. 1.94 Å), probably due to the steric repulsion between the *tert*-butyl groups on the biphenyl ligands, which suggests severe steric congestion around the central germanium atom of **1**. The theoretical structure of **1**, optimized at the B3PW91-D3(BJ)/6-311G(2d,p) level of theory, exhibits slightly longer Ge–C bonds (ca. 1.96–1.97 Å) relative to those of **III** (ca. 1.94 Å) when calculated at the same level of theory.

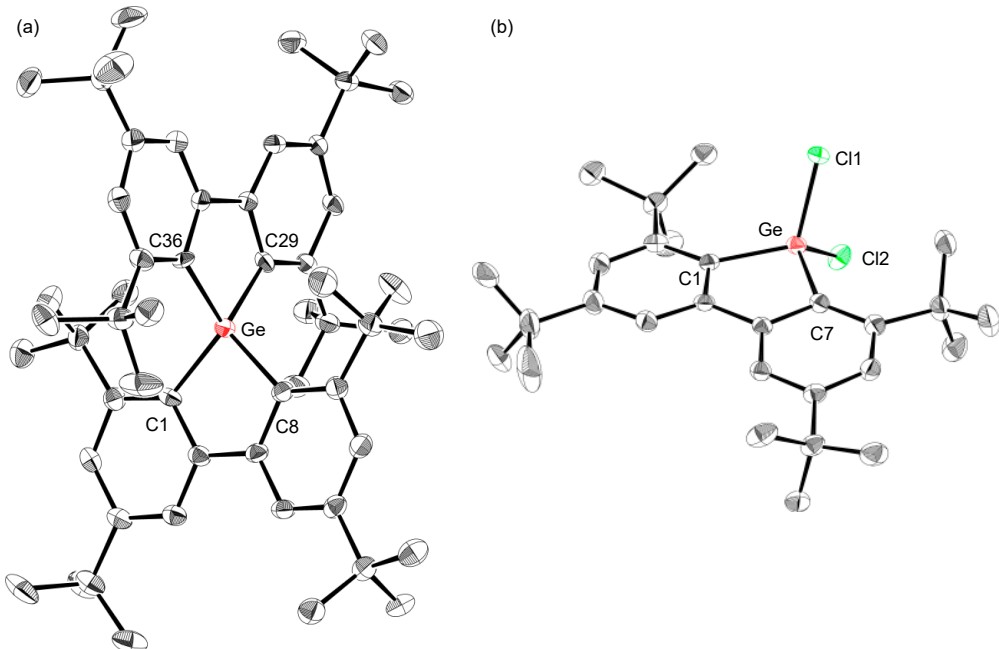

**Figure 2.** Molecular structures of (**a**) spirogermabifluorene **1** and (**b**) dichlorogermane **5** with thermal ellipsoids at 50% probability. Selected bond lengths (Å) and angles (°): (a) **1**, Ge–C1, 1.972(5), Ge–C8, 1.971(5), Ge–C29, 1.969(5), Ge–C36, 1.974(5), C1–Ge–C8, 91.8(2), C1–Ge–C36, 119.4(2), C8–Ge–C29, 120.9(2), C29–Ge–C36, 91.3(2); (b) **5**, Ge–Cl1, 2.1838(9), Ge–Cl2, 2.1791(9), Ge–C1, 1.936(3), Ge–C7, 1.935(3), C1–Ge–C7, 95.0(2), and Cl1–Ge–Cl2, 105.71(4).

The molecular structures of spirogermabifluorene **1** and dichlorogermane **5** determined by single-crystal X-ray diffraction analysis are shown in Figure 2 [32]. The Ge–C(biphenyl) bonds in **1** (1.974(5), 1.969(5), 1.972(5), 1.971(5) Å) are slightly longer than those of less hindered spirogermabifluorene **III** (1.938(2), 1.945(2) Å) [20] and **5** (ca. 1.94 Å), probably due to the steric repulsion between the *tert*-butyl groups on the biphenyl ligands, which suggests severe steric congestion around the central germanium atom of **1**. The theoretical structure of **1**, optimized at the B3PW91-D3(BJ)/6-311G(2d,p) level of theory, exhibits slightly longer Ge–C bonds (ca. 1.96-1.97 Å) relative to those of **III** (ca. 1.94 Å) when calculated at the same level of theory.

The electron affinity of the spirogermabifluorene was investigated based on theoretical calculations and electrochemical measurements. The theoretically calculated LUMO levels (eV) and electron affinity (EA in eV) values of Si- and Ge-spirobifluorenes **II** and **III**, and those of **1** and its Si analogue **10** are summarized in Figure 3; all calculations were carried out at the B3PW91-D3(BJ)/6-311G(2d,p) level of theory. Spirogermabifluorene **1** exhibits considerable spiroconjugation, which can be expected to lower the LUMO level, similar to the case of spirosilabifluorenes [4–10]. Unexpectedly, in both compounds with biphenyl ligands and 3,3′,5,5′-tetra-*t*-butylbiphenyl ligands, the Si analogues (**10** and **II**) exhibit lower LUMO levels and higher electron affinities than the corresponding Ge analogues (**1** and **III**), probably due to the less effective spiroconjugation in spirogermabifluorenes relative to spirosilabifluorenes on account of the longer Ge–C bond length. Notably, the 3,3′,5,5′-tetra-*t*-butylbiphenyl ligands were found to augment the EAs for both Si and Ge relative to the corresponding parent molecules (**II** and **III**), albeit that the LUMO levels are also raised. Consequently, the electron affinity of the overcrowded spirogermabifluorene **1** can be expected to be comparable to that of spirosilabifluorene **II**.

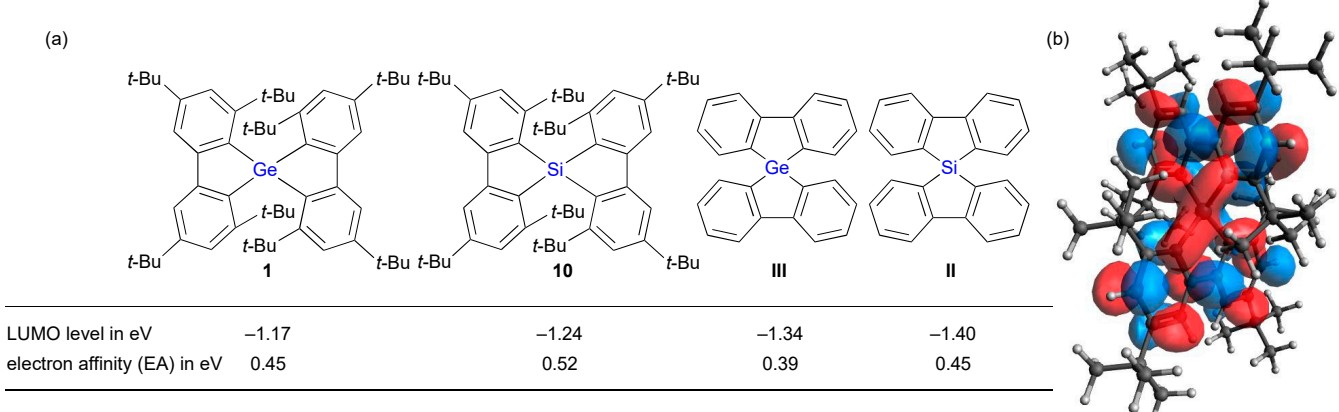

**Figure 3.** (**a**) Theoretically calculated LUMO levels and electron affinity (EA in eV) values for Si- and Ge-spirobifluorenes **II**, **III**, **1** and **10**. (**b**) LUMO of **1**; all calculations were carried out at the B3PW91-D3(BJ)/6-311G(2d,p) level of theory.

In contrast to spirosilabifluorene **II**, which is reduced by metals to give the corresponding anion-radical and dianion species as isolable compounds [33], spirogermabifluorene **III** decomposes with concomitant elimination of the biphenyl ligand upon reduction by metals [20]. Spirogermabifluorene **1** exhibited stable redox behavior in the electrochemical measurements, as shown in Figure 4. The cyclic voltammogram of **1** showed pseudoreversible redox couples, indicating appreciable redox stability under the applied conditions. While only one reduction wave was observed at $E_{pc} = -1.75$ V, the corresponding oxidations were observed separately at $E_{pa} = -1.11$ V and $-0.59$ V, albeit that the feature at $-1.11$ V was very weak and broadened. A combined consideration of the large difference between the $E_{pc}$ and $E_{pa}$ and the very broad peak observed in the differential pulse voltammogram of **1** suggested that **1** undergoes a two-electron reduction and that the oxidation of the generated dianion species is very slow due to the steric congestion afforded by the *tert*-butyl groups. In the oxidation region, **1** did not show any oxidation events under the same measurement conditions. In other words, **1** exhibits stable redox behavior in the reduction region, probably forming the corresponding dianion species [34].

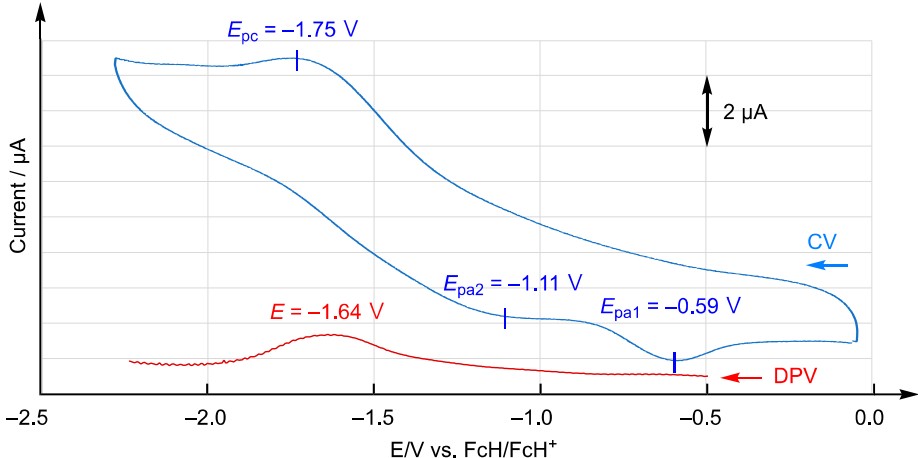

**Figure 4.** Cyclic (blue) and differential-pulse (red) voltammograms of spirogermabifluorene **1** in THF at $-40\ ^{\circ}$C ([**1**]: 1.0 mM; supporting electrolyte: 0.1 M [$n$Bu$_4$][PF$_6$]; scan rate: 70 mVs$^{-1}$).

## 3. Conclusions

A sterically hindered spirogermabifluorene (**1**) was successfully synthesized via the reaction of a dilithiated 3,3′,5,5′-tetra-*t*-butylbiphenyl ligand with an equimolar amount of GeCl$_2$·(dioxane). The sterically hindered biphenyl ligands afford **1** with high redox stability and increased electron affinity. We think that several families of spirogermabifluorenes

can be accessed using the synthetic methodology described in this paper, which may lead to new applications of spirogermabifluorenes on account of their unique optoelectronic features. The isolation and identification of the reduced product of spirogermabifluorene **1** and the synthesis of its Si-analogue are currently in progress in our laboratory.

## 4. Materials and Methods

### 4.1. General Information

All manipulations were carried out under an argon atmosphere using either Schlenk-line or glove-box techniques. All solvents were purified by standard methods. Residual trace amounts of water and oxygen in the solvents were thoroughly removed by bulb-to-bulb distillation from potassium mirror prior to use. $^1$H and $^{13}$C NMR spectra were measured on a JEOL ECZ-500R ($^1$H: 500 MHz; $^{13}$C: 126 MHz) or on a Bruker AVANCE-400 spectrometer ($^1$H: 400 MHz; $^{13}$C: 101 MHz). Signals arising from residual $C_6D_5H$ (7.15 ppm) in $C_6D_6$ or $CHCl_3$ (7.26 ppm) in $CDCl_3$ were used as the internal standards for the $^1$H NMR spectra; those of $C_6D_6$ (128.0 ppm) and $CDCl_3$ (77.0 ppm) were used for the $^{13}$C NMR spectra. High-resolution mass spectra were obtained from a JEOL JMS-T100LP (DART) or JMS-T100CS (APCI) mass spectrometer (DART). Gel-permeation chromatography (GPC) was performed on an LC-6AD (Shimadzu Corp., Kyoto, Japan) equipped with JAIGEL-1H and 2H (Japan Analytical Industry Co., Ltd., Tokyo, Japan) columns using toluene as the eluent. All melting points were determined on a Büchi Melting Point Apparatus M-565 and are uncorrected. 1-Bromo-3,5-di-*tert*-butylbenzene was generously donated by MANAC Inc., Tokyo, Japan.

### 4.2. Synthesis of Dichlorogermane **5**

2,2′-Dibromo-3,3′,5,5′-tetra-*t*-butylbiphenyl (1.07 g, 2.00 mmol)[26] was dissolved in $Et_2O$ (20 mL) and cooled to $-80$ °C. *n*-Butyllithium (1.64 M in hexane, 2.50 mL, 4.10 mmol) was slowly added at $-80$ °C. After stirring at $-80$ °C for 1 h, the reaction mixture was slowly warmed to 0 °C, where it was stirred for 2 h. The reaction mixture was then slowly added to a solution of $GeCl_4$ (0.40 M in $Et_2O$, 25.0 mL, 10.0 mmol) at $-80$ °C, and the resulting mixture was stirred for 1 h at the same temperature. Then, the reaction mixture was slowly warmed to room temperature and stirred for 3 h. After the removal of the volatiles under reduced pressure, LiCl (8.68 g, 200 mmol) and THF (200 mL) was added to the residue. After stirring at room temperature for 6 h, the volatiles were removed under reduced pressure. The residue was dissolved in hexane and filtered through a pad of Celite. The removal of the volatiles under reduced pressure gave a colorless solid. Recrystallization from hexane afforded **5** (0.637 g, 1.22 mmol, 61%) as colorless crystals. Mp. 139–142 °C. $^1$H NMR (400 MHz, $C_6D_6$) δ 7.94 (d, $^4J$ = 1.7, 2H), 7.66 (d, $^4J$ = 1.6, 2H), 1.64 (s, 18H), 1.26 (s, 18H); $^{13}$C NMR (100 MHz, $C_6D_6$) δ 156.9 (4°), 155.6 (4°), 143.8 (4°), 126.0 (CH), 125.4 (4°), 117.2 (CH), 38.2 (4°), 35.3 (4°), 32.9 ($CH_3$), 31.2 ($CH_3$); HRMS(DART-TOF), *m/z*: Found: 521.1816 ([M+H]$^+$), Calcd. for $C_{28}H_{41}{}^{70}Ge^{37}Cl_2$ ([M+H]$^+$): 521.1795. Anal. calcd for $C_{28}H_{40}GeCl_2$+0.2($C_6H_{14}$(hexane)): C, 65.26; H, 8.03. found: C, 65.64; H, 7.90.

### 4.3. Synthesis of Spirogermabifluorene **1**

2,2′-Dibromo-3,3′,5,5′-tetra-*t*-butylbiphenyl (1.07 g, 2.00 mmol) was dissolved in $Et_2O$ (20 mL) and cooled to $-80$ °C. *n*-Butyllithium (1.64 M in hexane, 2.50 mL, 4.10 mmol) was slowly added at $-80$ °C. After stirring at $-80$ °C for 1 h, the reaction mixture was slowly warmed to 0 °C, where it was stirred for 2 h. $GeCl_2$·(dioxane) (0.463 g, 2.00 mmol) was added to the reaction mixture at 0 °C in one portion. The reaction mixture was slowly warmed to room temperature and stirred for 12 h to afford an orange suspension. After the removal of the volatiles under reduced pressure in air, the residue was dissolved in $CH_2Cl_2$ and filtered through a pad of Celite. Removal of the volatiles under reduced pressure gave a yellow solid. The crude product was purified by GPC (toluene) to afford **1** (0.179 g, 2.17 mmol, 22%) as a colorless solid. Mp. 114–118 °C. $^1$H NMR (400 MHz, $C_6D_6$) δ 8.37 (d, $^4J$ = 1.8, 2H), 7.68 (d, $^4J$ = 1.7, 2H), 1.39 (s, 18H), 1.16 (s, 18H); $^{13}$C NMR (100 MHz,

$C_6D_6$) δ 155.5 (4°), 152.5 (4°), 148.3 (4°), 135.3 (4°), 125.4 (CH), 117.9 (CH), 37.9 (4°), 35.1 (4°), 32.4 ($CH_3$), 31.5 ($CH_3$); HRMS (DART-TOF), *m/z*: Found: 827.5593 ([M+H]⁺), Calcd. for $C_{56}H_{81}{}^{74}Ge$ ([M+H]⁺): 827.5565.

### 4.4. X-ray Crystallographic Analysis of Spirogermabifluorene 1 and Dichlorogermane 5

Single crystals of **1** and **5** were obtained by recrystallization from hexane. Intensity data for **1** and **5** were collected on a RIGAKU Saturn70 CCD(system) with VariMax Mo Optics using Mo-Kα radiation (λ = 0.71073 Å), while the preliminary data were collected at the BL02B1 beamline of SPring-8 (proposal numbers: 2018A1167, 2018B1668, 2018B1179, 2019A1057, 2019A1677, 2019B1129, 2019B1784, 2020A1056, 2020A1644, 2020A1650, 2020A0834) on a PILATUS3 X CdTe 1M camera using synchrotron radiation (λ = 0.4148 Å). The structures were solved using SHELXT-2014 and refined by a full-matrix least-squares method on $F^2$ for all reflections using SHELXL-2014. All nonhydrogen atoms were refined anisotropically, and the positions of all hydrogen atoms were calculated geometrically and refined as riding models. Supplementary crystallographic data were deposited at the Cambridge Crystallographic Data Centre (CCDC) under deposition numbers CCDC-2106926 (**1**) and CCDC-2106927 (**5**) and can be obtained free of charge via www.ccdc.cam.ac.uk/data_request.cif.

### 4.5. Electrochemical Measurements of Spirogermabifluorene 1

Cyclic and differential-pulse voltammograms were recorded on an ALS 1140A potentiostat/galvanostat using Pt wire electrodes under an argon atmosphere in custom-tailored glassware. Voltammograms were recorded at −40 °C on THF solutions ([analyte]: 1.0 mM; supporting electrolyte: 0.1 M [$n$Bu$_4$][PF$_6$]) using a variety of scan rates.

**Supplementary Materials:** The following are available online at https://www.mdpi.com/article/10.3390/inorganics9100075/s1. NMR spectral data of **1** and **5** (Figures S1–S4), Cacltatio details, UV/vis spectrum of intermediate **6** (Figure S5), and Crystallographic data for **1** and **5** (Table S1), crystallographic data (CIF), and theoretically optimized coordinates (xyz) are available in the Supplementary Materials.

**Author Contributions:** Conceptualization: T.S.; resources and funding acquisition: S.M. and T.S.; experiments: S.M. and K.N.; writing—original draft preparation: S.M., T.S.; writing—review and editing: S.M., K.N. and T.S.; project administration: T.S. All authors have read and agreed to the published version of the manuscript.

**Funding:** This research was funded by JSPS KAKENHI grants 19H02705 and 19K22188, as well as by 21K14607 from MEXT (Japan), by the University of Tsukuba Basic Research Support Program Type B, by the Collaborative Research Program of the Institute for Chemical Research at Kyoto University (2018-22), by the TOBE MAKI Scholarship Foundation (20-JA-014) and by JST CREST grant JPMJCR19R4.

**Institutional Review Board Statement:** Not applicable.

**Informed Consent Statement:** Not applicable.

**Data Availability Statement:** Data available in a publicly accessible repository.

**Acknowledgments:** We acknowledge the generous assistance of the Research Equipment Sharing Center at Nagoya City University, and the Supercomputer Laboratory in the Institute for Chemical Research of Kyoto University for the resources used. We would like to thank Shigehiro Yamaguchi, Makoto Yamashita, Kenichiro Itami, and Hiroshi Shinokubo (Nagoya University) for their kind support with the experimental work, and Toshiaki Noda and Hideko Natsume (Nagoya University) for the expert manufacturing of custom-tailored glassware. Moreover, we would like to express our appreciation to MANAC Inc. for a kind gift of 1-bromo-3,5-ditert-butylbenzene.

**Conflicts of Interest:** The funders had no role in the design of the study, in the collection, analysis, or interpretation of data, in the writing of the manuscript, or in the decision to publish the results.

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
