# Peer review of "Facile Synthesis and Redox Behavior of an Overcrowded Spirogermabifluorene"

_inorganics, doi:10.3390/inorganics9100075_

Round 1
Reviewer 1 Report
The paper of Takahiro Sasamori and co-authors is an interesting fundamental work on synthesis new spirogermabifluorene, determining its crystal structure and redox behaviour using Cyclic voltammetry. The problem authors were faced with was unworking classic method of synthesizing germanium compounds using reaction of GeCl4 with 2,2’-dilithio-3,3’,5,5’-tetra-t-butylbiphenyl due to steric hindrance. Authors proposed new scheme to obtain spirogermabifluorene through GeCl2·(dioxane) and 2,2’-dilithio-3,3’,5,5’-tetra-t-butylbiphenyl reaction and tried to find out its mechanism. The current work seems interesting to me and I recommend it to publish in Inorganics.
But I have some questions:
- Did you try to treat dichlorogermane 5 with magnesium in toluene? If your supposed mechanism on Scheme 3 is right you can get product 1 with higher yield, I think. Also, during generation the germylene you can fix/stabilize it with tolane for example like in Tsuyoshi Kato’s work: 10.1002/anie.201105639, 10.1002/anie.201201581. Maybe you will success in stabilizing 6B form.
- Did you find the Ge(0) in reaction mixture according to supposed mechanism on scheme 3?
Line 62: 2,2’-diblithio…
Author Response
Dear the reviewer,
Thank you very much for your kind e-mail regarding the results of the reviewing of our manuscript entitled "Facile Synthesis and Redox Behavior of an Overcrowded Spirogermabifluorene" by Shogo Morisako, Kohsuke Noro, and Takahiro Sasamori submitted as an article for the special issue ‘Redox-Active Ligand Complexes’ in the coordination chemistry section of Inorganics. We are pleased to see the positive comments, and are thus re-submitting the manuscript, which contains modifications in accordance with the suggestions of the reviewers. Our reply to the reviewers is attached. We hope that the revised manuscript is now acceptable for the publication in this special issue of Inorganics.
Yours sincerely,
Takahiro Sasamori

Reviewer 2 Report
The scientific content of the ms. describes the synthesis and characterization of a spirogermabifluorene and its dichlorogermane analogue. The main interest in the chemistry and reactivity of spirobifluorenes arises from their photophysical properties, which can be used for applications in organic light-emitting diodes and dye-sensitized solar cells. Organometallic analogs of spirobifluorene, having a central sp3-hybridized heteroatom, also demonstrate potentially useful optoelectronic properties and electron transport effects. Various compounds with a spirosilabifluorene core have been prepared and their fluorescent properties studied. On the other hand spirogermabifluorene compounds are scarce and the number of corresponding reports is limited. Thus, this ms deserves acceptance in Inorganics as it describes the preparation and characterization of two novel compounds in the area of spirogermabifluorenes. The paper-when published-will attract the interest of researchers working in the general areas of inorganic and organometallic chemistry of spiroconjugated compounds. Also I anticipate that the article will receive a respectable number of citations in the future.
Specific revision points/comments/suggestions raised from this ms are:
- The English of the ms should be improved. Phrases such as ‘’…tethered to a central atom…’’ and ‘’…furnished the corresponding spirogermabifluorene…’’, should be written with proper terminology such as tethered=joined by and furnished= lead to the isolation.
- On line 63 the correct name of the compound 4 is 2,2΄-dilithio-3,3΄,5,5΄-tetra-t-butylbiphenyl instead of 2,2΄-dibllithio-3,3΄,5,5΄-tetra-t-butylbiphenyl. Also, on the same line the number in parenthesis should be changed to the corresponding compound 4.
- The authors claim that the possible intermediates 6A and 6B were identified through UV-Vis spectroscopy but the corresponding spectra are missing from the ms. and the Supporting Information. I suggest that the UV-Vis spectra should be placed in the Supplementary Information and cited in the main ms.
- The second paragraph in the section “Results and Discussion” and Schemes 1 and 2 that describe the formation and isolation of compound 1 are rather confusing. As far as I can understand the compound 1 has been prepared in an one pot reaction through two steps. The first step is the in situ formation of dilithium compound 4 (depicted in Scheme 1) and the second step includes the reaction of GeCl2∙(dioxane) with the in situ prepared dilithium compound 4 that led to the formation of 1, through the formation of the two intermediates (6A and 6B) after cooling the reaction system to room temperature. In Scheme 2, in the reaction scheme there is a label x2 instead of RT. It would be preferable to place in one scheme only the reaction systems that lead to the formation of the structurally characterized compounds (1 and 5) and in another scheme the failed attempts and the isolation of the dibromo compound 7. Also, the characterization data for the prepared compound 7 should be placed in the “Material and Method” section and not in the reference section. Additionally, the 1H NMR and HRMS spectra should be placed in the Supplementary Information and cited in the main text of the ms.
- A table with all the crystallographic data should be placed in the Supplementary Information. Also, in reference 14 that includes the crystallographic data of the two prepared compounds (1 and 5) the numbers of the corresponding compounds should be the same with those referred to the main text of the ms (5 instead of 2, line 341).
- In Figure 2, according to the cif file of the compound 1, the C atoms attached to the Ge metal center are C1, C8, C29, and C36 and not C1,C2, C3 and C4. Also, for the compound 5 the corresponding C atoms are C1 and C7, instead of C1 and C2. Please fix the corresponding labels in Figure 2, and change them also in the label of Figure 2 and in the main text in the section where the distances are described (line 128).
- Please, revise the references in the main text according to the instruction of the journal ([1]. Instead of .[1]). Also, it would be preferable to put every reference cited in the text independently and not merged in the same citation.
- In the Conclusion Section, the perspectives of this work should be outlined.
Author Response

(The authors gave the same response as above.)

Reviewer 3 Report
This paper describes the syntheses, crystal structures and electrochemical properties of a spirogermabifluorene and a related dichlorogermane with 3,3’,5,5’-tetra-t-butylbiphenyl ligand. The estimation of the synthetic route and electrochemical properties are supported by theoretical calculations. This is a carefully done study and the findings are interesting. Thus, this paper is worth publishing in Inorganics with minor revisions. Additional comments are listed below.
1) page 2, line 62: diblithio --> dilithio
2) page 2, line 64: the thus in-situ-generated --> the in-situ-generated
Author Response

(The authors gave the same response as above.)
